# Effect of Chloride and Ferrous Ions on Improving Copper Leaching from Black Copper Ores

Rossana Sepúlveda [1,*], Melissa Martínez [1,2], Pía Hernández [3], Alexis Guzmán [1] and Jonathan Castillo [1]

1. Departamento de Ingeniería en Metalurgia, Universidad de Atacama, Av. Copayapu 485, Copiapó 1530000, Chile; nastassia.mmg@gmail.com (M.M.); alexis.guzman@uda.cl (A.G.); jonathan.castillo@uda.cl (J.C.)
2. División Salvador, CODELCO, Bernardo O'Higgins 103, Diego de Almagro 1500000, Chile
3. Departamento de Ingeniería Química y Procesos de Minerales, Universidad de Antofagasta, Av. Angamos 601, Antofagasta 1270300, Chile; pia.hernandez@uantof.cl
* Correspondence: rossana.sepulveda@uda.cl; Tel.: +56-52-2-255620

**Abstract:** Although the importance of black copper ores is well established, many topics remain to be studied. This manuscript describes the efforts to improve copper recovery from refractory ores on a pilot scale. Preliminary tests provided the water dose to form a stable and compact agglomerate of 89 L per ton of ore and an acid dosage of 40 kg per ton of ore. The column leaching method resulted in an effluent with 1.63 g/L Cu, a pH of 1.04, and a redox potential of 675 mV (average). Despite the similarities observed in the results, reductant agents were essential to dissolve the $MnO_4$ present in the black copper ore. Thus, the best Mn extraction results were 67.7% using the $MnO_2$:$FeSO_4$ ratio of 1:3. Additionally, the kinetics of leaching are slower than in an industrial operation. The copper ore under study required more than 65 days of leaching to reach the maximum copper recovery. Finally, the total recovery of copper (including washing and resting time) indicates that the maximum value was reached for ore pretreated with NaCl, $H_2SO_4$, and $FeSO_4$, concluding that the use of agents favored leaching.

**Keywords:** black copper ores; copper recovery; refractory ores; pilot scale; column leaching method

## 1. Introduction

Black copper ores are part of the mineralogy of the so-called exotic deposits. Exotic deposits were formed from the oxidation and leaching of primary porphyry copper deposits (secondary enrichment) [1]. The formation of exotic Cu deposits is a very complex phenomenon, including tectonic processes, climate, erosion, initial orebody, adjoining rocks, water table level, the evolution of pH/Eh of Cu-bearing aqueous solutions, and bacterial activity. Favorable conditions and large amounts of copper sulfides cause a new enrichment to develop from the original porphyry; these conditions occur in some parts of the world, such as northern Chile and Spain [2–4]. The paragenesis of exotic deposits has been studied extensively using experimental and modeling data. Dold [5] reported the result of paragenesis modeling and proposed that the mechanisms of exotic bodies are the water–rock interaction process of acid rock drainage with the specific host-rock mineralogy. The acid rock drainage with elevated $Cl^-$ and Si concentrations mobilized cupric ions in low-pH conditions. These acid solutions contact the neutralizing bedrock, resulting in the over-saturation and subsequent precipitation of complex mineralogical species. Different information exists in the literature regarding the characterization of exotic Cu deposits. Much of the efforts are on modern technique characterization because traditional optical microscopy is not accurate enough to differentiate Cu–Fe–Mn species. Exotic Cu deposits have been extensively studied using isotope techniques, inductively coupled plasma mass spectroscopy, X-ray diffraction, and scanning electron microscopy [6,7]. These papers report the geological age of the formation of the exotic deposits and explain the mineralization of

the deposits in the case of copper. Helle et al. report that the copper grades of the samples studied range from 0.3 to 3.2%, iron from 2 to 4.3%, and manganese from 0.08 to 2.7%. The mineralogical species identified are chrysocolla, atacamite, copper clays, and black coppers, among others. Finally, multi-step leaching tests showed copper recoveries between 56–93%, iron between 7–43%, and manganese between 17–75%. New technologies make it possible to estimate the chronology, elemental composition, and different mineralogical species. Kidder et al. reported exotic mineralization through a complete analysis of groundwater samples, utilizing spectroscopy, ion chromatography, and stable isotopes of Cu, Mo, S, and $^{87}Sr/^{86}Sr$ analysis [8].

Historically, the term "black copper" has been employed in mining terminology to identify copper ores of black color with a variable chemical composition and an undefined crystalline structure. Black copper ores are a family of amorphous or poorly crystalline Cu-bearing oxides. These kinds of minerals have a variable chemical composition of Cu, Mn, Fe, Si, and Cl. The morphology and mineralogy exhibit nanometer-sized cylindrical structures of atacamite, quartz, pseudomalachite, dioptase, neotocite, gypsum, paratacamite, and melanothallite [9,10]. On the topic of black copper ores, the available literature recurrently mentions two species: copper pitch and copper wad [11]. Copper wad is the generic term for copper-bearing Mn oxyhydrates that are difficult to characterize accurately. Exhaustive analysis of copper wad samples shows a complex mixture of poor-crystalline chrysocolla, atacamite, libethenite, and gypsum. Mote et al. describe copper wads as complex structures formed by Cu-rich K-bearing Mn oxyhydrates with cryptomelane and birnessite [12]. The description of copper pitch varies from copper wad because of the association with chrysocolla; in this case, copper wad is also called "black chrysocolla" [13]. Despite efforts to characterize black copper ores, Pincheira et al. propose no significant differences between copper pitch and copper wad species and therefore consider that they should be called black copper silicates [14].

The comprehensive geochemical analysis of exotic Cu deposits has been very beneficial in understanding the complex mineral associations. However, the current literature needs more reports that connect knowledge of hydrometallurgical processing and the geological study of exotic deposits. The processing of exotic deposits by leaching requires a thorough understanding of the mineralogy, as metallurgical performance is often poor. Black copper is shown as a species refractory to leaching and presenting occlusion textures. Therefore, efforts are focused on exhaustively determining the mineralogy of the ore, followed by metallurgical tests to predict and optimize copper dilutions [14,15]. Pincheira et al. [14] state that the mineralogical species associated with copper are mainly chrysocolla and atacamite, in addition to an amorphous gel rich in silica, copper, iron, and manganese (black copper), which is described as the most refractory species to leaching. The researchers conclude that the relationship between Cu–Mn–Fe–Al–Si is complex, where metallurgical copper recoveries through leaching can reach values close to 100% for samples with a low presence of complex minerals and values close to 80% for highly occluded samples. According to Pinget et al. [10], the exotic body located downstream of the Chuquicamata porphyry deposit has copper grades ranging from 0.7 to 1.5 Cu wt.%. The copper mineralization is mainly represented by chrysocolla, atacamite, and copper wad, along with pseudomalachite, paracoquimbite, lebethenite, and sampleite. Helle et al. analyzed two Mina Sur, Chuquicamata (Chile), samples. The modal analysis of the samples showed that sample 1 contained 13% chrysocolla, 1% copper pitch, and 1% copper wad, while sample 2 contained 1% atacamite and 1% copper wad. The leaching recovery was 93% for sample 1 and 83% for sample 2. It is important to note that these results correspond to tests conducted under ideal laboratory conditions.

In this respect, research in this area has shown that low redox potential favors the dissolution of manganese, increasing the copper extraction efficiency for refractory ores [16,17]. In studying exotic minerals, it is crucial to consider the contribution of manganese oxide, which can significantly impact copper extraction efficiency. Additionally, the chloride medium is crucial in improving the dissolution of the refractory ores or concentrates.

Velásquez-Yévenes and Lasnibat have researched to improve the copper dissolution efficiency of ores with black copper contents (11 to 18% copper wad) using organic reducing agents and chlorides. They found that adding 50 g/L of NaCl and 60 g/L of cane molasses to the leaching solution enhanced the dissolution of the manganese and exotic copper ore. As a result of their experiments, they achieved close to 100% copper recoveries. In the study of Toro et al., the researchers mixed chalcopyrite with manganese oxide to determine its effect on copper leaching. The best results of this investigation were obtained when working at $MnO_2$:$CuFeS_2$ ratios of 5:1 and a concentration of 1 mol/L of $H_2SO_4$ at 25 °C, which resulted in a promising 77% copper recovery. These findings highlight the importance of considering the influence of manganese and chloride concentrations in the leaching process of exotic minerals. In contrast, the addition of oxidant agents does not benefit copper recovery. This fact is attributable to an increase in redox potential [18,19]. For example, Pérez et al. proposed focusing on dissolving $MnO_2$ in black copper using reducing agents. The researchers compared the use of ferrous ions and tailings as reducing agents. The study shows that ferrous ions were the best reducing agent to dissolve $MnO_2$ in black copper. The best working condition was a $Fe^{2+}$:black copper ratio of 2:1 and one mol/L of sulfuric acid [20]. Quezada et al. demonstrate high copper recoveries using a low redox potential. For example, copper recoveries are nearly 90% at 450 mV versus 65% at 600 mV [21].

The addition of chlorides in leaching has been widely studied to improve the leaching performance of complex ores and copper concentrates [22–25]. The utilization of an acidic aqueous medium containing chlorides has been extensively documented in the leaching and bioleaching processes of copper ores, such as chalcocite, chalcopyrite, and other low-grade copper sulfide minerals. In this type of medium (chloride), there are several aspects to be studied that make the chemistry of the process more complex; for example, the effects of ferric/ferrous ions, cupric/cuprous ions, elemental sulfur, and passivation ions, among others. The mechanism involved in the chloride medium of chalcopyrite is shown below [26].

$$CuFeS_2 + 3[CuCl]^+ + 5Cl^- \rightarrow 4CuCl + Fe^{2+} + 2S$$

$$CuFeS_2 \rightarrow CuS + FeS$$

$$CuS + CuCl_2 \rightarrow 2CuCl + S$$

One of the main problems in the leaching of refractory ores is the formation of a layer that inhibits the contact of the ore with the aqueous medium. To reduce the problem described above, Castellon and Taboada reported using high chloride and iodide ions in copper leaching from chalcopyrite. In this system, copper extraction reached 45% within 96 h, while at 216 h, it reached an extraction of close to 70% copper [27]. In the case of bioleaching, Bakhshoude and co-workers study bacteria such as *Sulfolobus acidocalarius* that are adapted to the chloride environment. The copper concentrate at pH 1.5, 1% of solids, and sodium chloride concentrations of 0.5 and 1.0 M is bioleaching at close to 100% after 21 days. Under the same conditions without microorganisms, copper dissolution is about 62% [28]. In the same vein, the addition of chlorides has also been studied to dissolve black coppers [29–32]. Thus, Torres and co-workers [33] published the use of chloride and ferrous ions to leach black copper ore. The result shows a poor copper recovery with only NaCl and sulfuric acid. In contrast, the best result for leaching tests was a $Fe^{2+}$:$MnO_2$ ratio of 1:1 and 20 kg/t of sulfuric acid.

This research assessed the curing and leaching performance of copper ore with black copper mineralogical species. We used the pilot-scale bottle test and column leaching methods to obtain results emphasizing industrial-scale application. Thus, the variables studied were NaCl addition, curing time, and reducing agent (ferrous ions).

## 2. Materials and Methods

### 2.1. Black Copper Ore Characterization

The oxidized Cu mineral used in the metallurgical tests was extracted from the San Antonio mine stock, Salvador Division (CODELCO Chile, Salvador, Chile). The material collected has the same physical characteristics as the industrial conditions. The particle size was 100% under 12.7 mm. The chemical composition of the ore was determined by atomic spectrometry (SpectrAA 220, Varian, Palo Alto, CA, USA). The results of chemical analyses are shown in Table 1. The total sulfuric acid consumption was 40 kg per ton of ore.

**Table 1.** Chemical composition of the copper ore.

| Total Copper (%) | Soluble Copper (%) | Fe (%) | Mn (%) | Mg (%) | Ca (%) | Al (%) |
|---|---|---|---|---|---|---|
| 0.71 | 0.33 | 5.37 | 0.46 | 0.10 | 0.62 | 0.21 |

Mineralogical analyses of the copper ore samples were performed using a Bruker D8 Advance X-ray diffractometer (XRD; Bruker, Billerica, MA, USA). The result indicated that the main crystalline phases present in the sample were quartz, orthoclase, and albite. The sample's content of poorly crystalline or amorphous compounds was 65.5%. The complete mineralogical composition of the ore samples (Table 2) was analyzed using automated mineralogy techniques by scanning electron microscopy (QEMSCAN®, Carl Zeiss, Oberkochen, Germany). The gangue is mainly hornblende, plagioclase, feldspars, and quartz. The copper-bearing oxide minerals in the ore consist mainly of copper wad, copper-bearing phyllosilicates, copper-bearing Fe oxide/hydroxides, and chrysocolla.

**Table 2.** Mineralogical composition of the ore sample.

| Mineral | Chemical Formula | Mass (%) |
|---|---|---|
| Chrysocolla | $(Cu,Al)_2H_2Si_2O_5(OH)_4 \cdot n(H_2O)$ | 0.29 |
| Cu-bearing Phyllosilicates | $(Cu,X,Y)_a(Si_2O_5)_b(OH)_c$ | 5.45 |
| Cu-bearing Fe Oxide/Hydroxides | $CuxFe^{3+}O(OH)$ | 0.52 |
| Cu-bearing Wad | $(Cu,Al,Mn,Fe)_xO_y$ | 2.08 |
| Other Cu Minerals | $Cu_xO_y(OH)_z$ | 0.02 |
| Pyrite | $FeS_2$ | 0.01 |
| Fe Oxides/Hydroxides | $Fe_xO_y(OH)_z$ | 1.75 |
| Sulfates | $CaSO_4 \cdot 2H_2O/CaSO_4$ | 0.07 |
| Carbonates | $CaCO_3/MgCO_3$ | 5.44 |
| Quartz | $SiO_2$ | 22.36 |
| K-Feldspars (Orthoclase, Anorthoclase) | $KAlSi_3O_8$ | 15.81 |
| Ca, Na-Feldspars (Plagioclase Series) | $(Ca,Na)Al_2Si_2O_8$ | 21.71 |
| Kaolinite Group | $Al_2Si_2O_5(OH)_4$ | 1.26 |
| Muscovite/Sericite/Illite/Phengite | $KAl_2(Si_3Al)O_{10}(OH,F)_2/(K,H_3O)(Al,Mg,Fe)_2(Si,Al)_4O_{10}[(OH)_2,(H_2O)]$ | 3.65 |
| Smectite Group (Montmorillonite, Nontronite) | $(Na,Ca,Fe)_x(Al,Mg)_2Si_4O_{10}(OH)_2 \cdot n(H_2O)$ | 2.21 |
| Biotite/Phlogopite | $K(Mg,Fe^{2+})_3[AlSi_3O_{10}(OH,F)_2$ | 0.74 |
| Chlorite Group | $(Fe^{2+},Mg,Fe^{3+})_5Al(Si_3Al)O_{10}(OH,O)_8$ | 1.51 |
| Hornblende | $Ca_2Mg_4Al_{0.75}Fe^{3+}{}_{0.25}(Si_7AlO_{22})(OH)_2$ | 9.46 |
| Epidote | $Ca_2(Fe,Al)Al_2(SiO_4)(Si_2O_7)O(OH)$ | 2.70 |
| Others | ----- | 2.97 |

### 2.2. Iso-pH Leaching Test Methods

The iso-pH test was performed by a bottle roll leach test (laboratory scale) to evaluate the suitability of the ore against the leaching process. This test aimed to obtain the maximum copper recovery and acid consumption. The bottle roll leach test was performed at room temperature (20 °C approx.) for 15 h, operated at 50 rpm rotational speed in polypropylene drums of 10 L. The iso-pH test used 1 kg of sample and 2 L of the aqueous solution. The

test ended when the pH did not change from 1.5, the ore stopped consuming acid, and the Cu recovery curve became asymptotic.

### 2.3. Curing Tests Methods

Three kilograms of copper ore were used in each curing test. The samples were manually mixed and agglomerated with the solution over an impermeable surface of high-density polyethylene (HDPE). After agglomeration, the ores were placed in plastic bags. The sulfuric acid dosage, resting time, and type of agent (oxidizing or reducing) were studied through curing tests. Five resting times, three dosages of sulfuric acid, and doses of reducing/oxidizing agents were investigated. The ferrous ion dosage was used under a $MnO_2:Fe^{2+}$ ratio of 1:1, 1:2, and 1:3. The sulfation tests used mineral ore, distilled water, sodium chloride, iron(II) sulfate heptahydrate, and sulfuric acid. All reagents are analytical grade and supplied by Merck (Rahway, NJ, USA).

The curing tests were extensive, and 240 tests were performed. In addition, three variables were analyzed at different levels, as shown in Table 3.

**Table 3.** Variables studied in the curing tests.

| Variable | Value | Unit |
|---|---|---|
| Curing time | 24, 48, 72, 96, 120 | h |
| Acid dosage | 40, 50, 60 | % of analytical acid consumption |
| NaCl | 10, 20, 30 | kg/t |
| $MnO_2:Fe^{2+}$ (as $FeSO_4$) | 1:1, 1:2, 1:3 | - |

### 2.4. Leaching Column Tests

The material was agglomerated and loaded into cylindrical columns made of HDPE material with a height of 1 m and a diameter of 315 mm (considering duplicates). Then, the pretreatment stage was carried out by adding water and acid according to the results obtained in the previous stage. In the case of the samples containing NaCl and $FeSO_4$, the reagents were added before the addition of water and acid to simulate the industrial process (mixed on a conveyor belt before the agglomerating drum). Once the pretreatment stage was completed, the agglomerates produced were placed in columns. Irrigation of the ore loaded in columns was carried out employing peristaltic pumps. The leaching column tests were performed at room temperature (approx. 20 °C). The columns were monitored daily. The experimental condition of the leaching tests is shown below in Table 4.

**Table 4.** Experimental conditions of the leaching tests.

| Column | Pretreatment |
|---|---|
| Column 1 (duplicate) | $H_2SO_4$ |
| Column 2 (duplicate) | $H_2SO_4$, and NaCl |
| Column 3 (duplicate) | $H_2SO_4$, and $FeSO_4$ |
| Column 4 (duplicate) | $H_2SO_4$, NaCl and $FeSO_4$ |

## 3. Results and Discussion

### 3.1. Iso-pH Leaching Test

First, it was determined that the ore was saturated by adding 11% water, which does not allow it to agglomerate, leaving a slurry that does not allow for loading of the leaching columns. Therefore, the optimum moisture value to avoid saturation of the sample was 89 L of water per ton of ore, achieving a bulk density of 1.42 g/cm$^3$.

The results of the iso-pH bottle test are shown in Figure 1. The result shows a maximum copper recovery of 41%. This result does not consider the addition of reducing agents; therefore, it shows the leaching performance under the current conditions of a leaching

plant. In addition, the results determined a sulfuric acid consumption of 33.4 kg per ton of ore.

**Figure 1.** Percentage of copper extraction and acid consumption (kg $H_2SO_4$/t).

After the preliminary tests were completed, the parameters for curing tests of copper ore with a high black oxide content were determined. Finally, the water dosage was set at 89 L/t of ore and an acid dosage of 40 kg of sulfuric acid per ton of ore.

### 3.2. Curing Tests

Table 5 illustrates the breakdown of copper sulfation according to curing tests. The statistical analysis made it possible to determine the factors' behavior independently. Thus, it was determined that 72 h of rest are adequate for the variables and levels under study. Furthermore, the acid dosage was set based on the previous iso-pH tests. Although the optimal time was 120 h of rest, under the industrial criteria, it is complex to implement in Salvador's operation. Thus, it is estimated that the most adequate time to work in an actual industrial condition will be 72 h, where a good copper recovery is achieved at a resting time according to a large hydrometallurgy plant. An increase in the resting time could be considered for possible projects and modifications to the hydrometallurgical plant. Finally, regarding the type of agent, the combination of 30 kg/t NaCl and 1:3 $MnO_2$:$FeSO_4$ is the most efficient when analyzing sulfation efficiency. In contrast, the interaction of all the variables under study implies that the best result for the sulfation tests would be achieved under conditions of 72 h of resting time and 30 kg/t NaCl + 1:2 $MnO_2$:$FeSO_4$.

**Table 5.** Result of curing tests for 60% of the analytical consumption of $H_2SO_4$ (24 kg of sulfuric acid per ton of ore) and three dose levels of $MnO_2$:$FeSO_4$ and NaCl.

| Time (h) | $MnO_2$:$FeSO_4$ | | | | | | | | |
|---|---|---|---|---|---|---|---|---|---|
| | 1:1 | | | 1:2 | | | 1:3 | | |
| | 10 kg/t NaCl | 20 kg/t NaCl | 30 kg/t NaCl | 10 kg/t NaCl | 20 kg/t NaCl | 30 kg/t NaCl | 10 kg/t NaCl | 20 kg/t NaCl | 30 kg/t NaCl |
| 24 | 14.7% | 17.7% | 11.3% | 15.6% | 14.9% | 24.2% | 10.8% | 15.9% | 23.1% |
| 48 | 23.3% | 20.4% | 19.4% | 21.8% | 28.9% | 23.6% | 23.3% | 22.5% | 22.6% |
| 72 | 28.6% | 24.1% | 21.1% | 25.1% | 26.3% | 29.6% | 25.1% | 26.7% | 23.0% |
| 96 | 19.5% | 18.2% | 25.4% | 19.0% | 19.3% | 26.8% | 23.6% | 22.6% | 27.0% |
| 120 | 21.7% | 17.5% | 18.0% | 21.9% | 26.0% | 20.3% | 18.7% | 21.1% | 35.6% |

Some authors agree that more emphasis should be given to the curing process to improve copper recovery from refractory ores. By enhancing the curing, for example, by adding chloride or nitrate ions, better copper recoveries will be obtained with less effort in the leaching stage [34].

The positive effect of pretreatment from refractory copper ores was reported by Cerda et al., who studied the leaching of primary copper sulfide in acid–chloride media. These researchers conclude that pretreatment is key to improving copper recovery. The researchers studied curing with 7, 20, and 40 days of resting time. The maximum copper recovery (93% Cu) is obtained with a pretreatment of 40 days and 90 kg of Cl-/t ore at 50 °C [31]. These studies aim to demonstrate the effect of curing over very long periods of resting time, but in practice, the resting time is much shorter at the industrial level. In this sense, Hernández and collaborators [30] studied curing with a 3-day resting time, where they obtained an increase from 26.8% copper recovery (without curing) to 64.7%. On the same topic, Velásquez-Yévenes et al. [35] reported the importance of rest time in curing. The report indicates that the high curing period increases the rate of dissolution of chalcopyrite. The curing time studies by Velazquez et al. were 15 to 80 days.

### 3.3. Leaching Column Tests

The leaching time of the columns was determined based on industrial operating parameters such as column height, leaching time, leach solution rate, and granulometry. As a result, the maximum leaching efficiency at the industrial level is reached in less than 65 days.

The copper recovery (%) by column is shown in Figure 2. The copper recovery reached between 45 and 50%. In fact, on day 65 of the leaching cycle, the copper recovery showed an accelerated dissolution of the metal. This fact means that the irrigation should have continued. In addition, it is observed that after day 45, there is an increase of 1% in copper recovery due to intermittent irrigation (similar to the industrial irrigation cycle). The mass balance to determine the total copper recovery (after the leaching cycle) includes washing the tailings with water for two days and resting for another two days. Thus, the total copper recovery obtained by columns 1 to 4 was 54.4, 55.0, 56.9, and 57.8%, respectively. The effluent pregnant leach solutions (PLS) on the last day of the leaching cycle are shown in Table 6. The solutions are similar for the four columns, and no significant differences are observed between the columns under study.

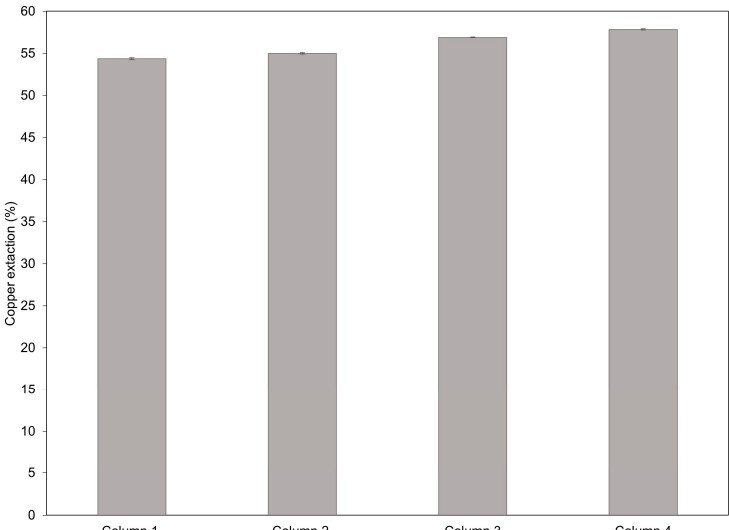

**Figure 2.** Cu recovery by leaching cycle solutions. Column 1: Pretreatment with $H_2SO_4$ and leaching; Column 2: Pretreatment with $H_2SO_4 + NaCl$ and leaching; Column 3: Pretreatment with $H_2SO_4 + FeSO_4$ and leaching; and Column 4: Pretreatment with $H_2SO_4 + NaCl + FeSO_4$ and leaching.

**Table 6.** PLS characteristics of the last day of the leaching cycle.

| Column | $Cu^{2+}$ (g/L) | $Fe^{2+}$ (g/L) | $Fe^{3+}$ (g/L) | $Mn^{2+}$ (g/L) | $SO_4^{2-}$ (g/L) | pH |
|---|---|---|---|---|---|---|
| 1 | 1.63 | 1.10 | 12.77 | 3.63 | 129 | 1.03 |
| 2 | 1.50 | 1.50 | 12.19 | 3.47 | 187 | 1.00 |
| 3 | 1.63 | 1.20 | 13.34 | 3.75 | 186 | 1.08 |
| 4 | 1.50 | 1.30 | 12.47 | 3.62 | 187 | 1.05 |

The positive effect of reducing agents on the dissolution of some copper species is critical [36]. In this study, many reducing agents were not used because these are semi-industrial tests (pilot scale). However, other authors have lowered the redox potential to values below 500 mV to achieve higher copper dissolution in a laboratory context [24,37].

Figure 3 shows the recovery of manganese obtained in column tests. Reducing agents were essential to dissociate $MnO_2$ in an acidic medium. In fact, the best Mn extraction results were 67.7% obtained in column 3, using the $MnO_2$:$FeSO_4$ ratio of 1:3. Therefore, by adding $FeSO_4$ in the pretreatment stage, it was possible to dissolve a greater amount of Mn present in the initial mineral sample.

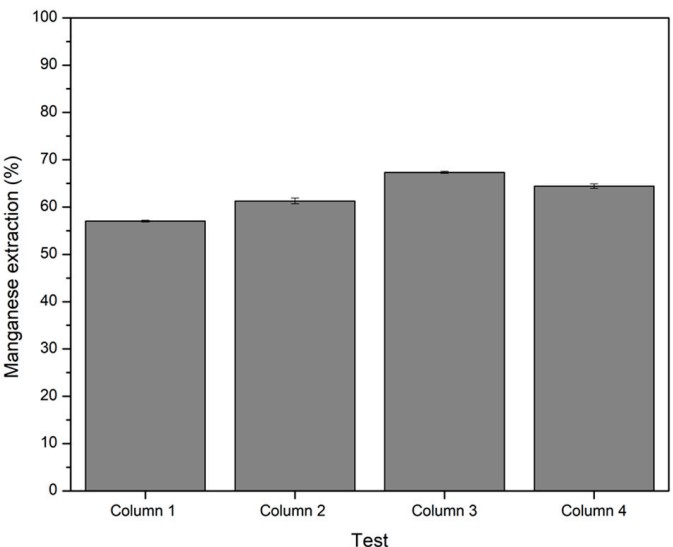

**Figure 3.** Manganese recovery in the column leaching tests. Column 1: Pretreatment with $H_2SO_4$ and leaching; Column 2: Pretreatment with $H_2SO_4$ + NaCl and leaching; Column 3: Pretreatment with $H_2SO_4$ + $FeSO_4$ and leaching; and Column 4: Pretreatment with $H_2SO_4$ + NaCl + $FeSO_4$ and leaching.

Regarding manganese dissolution, Perez et al. [20,37] suggest a simple mechanism for leaching a common manganese oxide such as pyrolusite (reaction 1). However, the mechanism is more complex for the dissolution of an exotic copper mineral in the presence of the ferrous ion (reaction 2). Finally, the leaching of mineralogical species containing manganese is predominant in copper recovery. Low oxidation potentials favor the dissolution of manganese by adding reducing agents [38].

$$2FeSO_{4(aq)} + 2H_2SO_{4(aq)} + MnO_{2(s)} = Fe_2(SO_4)_{3(s)} + 2H_2O_{(l)} + MnSO_{4(aq)}, \qquad (1)$$

$$CuO_xMnO_{2x}7H_2O_{(s)} + 3H_2SO_{4(aq)} + 2FeSO_{4(aq)} = Fe_2(SO_4)_{3(aq)} + MnSO_{4(aq)} + CuSO_{4(aq)} + 10H_2O_{(aq)}, \qquad (2)$$

The mechanism shown in reaction 3 changes in the presence of chloride. It is proposed that the $Cu^{2+}$/$Cu^+$ pair, stabilized by the effect of the chloride ion, plays an essential role in the dissolution of manganese [17]. The cuprous ion is not stable in aqueous media, but it can be stabilized in high chloride-containing media by forming chloride complexes. Thus, the

leaching shown above is controlled by the thermodynamic balance of the Cu(I)/Cu(II) complexes. This mechanism is more efficient than the Fe(II)/Fe(III) pair and has been used in several leaching studies of refractory copper ores using high chloride concentrations [26,39].

$$MnO_{2(s)} + 2CuCl_3^{2-}{}_{(aq)} + 4H^+{}_{(aq)} = Mn^{2+}{}_{(aq)} + 2CuCl^+{}_{(aq)} + 4Cl^-{}_{(aq)} + 2H_2O_{(aq)}, \quad (3)$$

*3.4. Analysis of the Residue of the Column Leaching Tests*

After the leaching cycle, the residue was sampled and analyzed to determine the chemical and mineralogical composition (AA and QEMSCAN). The grade of the residues was 0.27% Cu. The copper modal mineralogical species were phyllosilicate (0.17%), copper wad (0.05%), and goethite (0.02%). Regarding the main mineralogical species of the present study, the copper wad was dissolved by 73.3% in column 1, 73.0% in column 2, 75.0% in column 3, and 74.7% in column 4. These results allow us to deduce that $FeSO_4$ has a moderate effect on improving copper recovery.

This result is opposite to the reports of Benavente et al., who analyzed black copper leaching residues. The authors obtained high copper recoveries but very low manganese dissolution. This fact only leads to the conclusion that exotic copper ores are complex in their composition and response to leaching. There is no "kitchen recipe" for the beneficiation of exotic copper ores [18].

**4. Conclusions**

The present experimental study was designed to determine the effect of chloride and ferrous ions on improving copper leaching from black copper ores. The results suggest that:

- The best combination of parameters for the sulfation test was a rest time of 72 h, 60% of the analytical consumption of $H_2SO_4$, 30 kg/t of NaCl, and 1:3 $MnO_2$:$FeSO_4$. Reducing agents were essential to dissociate the $MnO_2$ present in the black copper ore. It was verified when the best Mn extraction was obtained using the $MnO_2$:$FeSO_4$ ratio of 1:3. Therefore, adding $FeSO_4$ in the pretreatment stage made it possible to dissolve a more significant amount of Mn.
- The maximum copper recovery in the column test was obtained for the sample pretreated with NaCl + $H_2SO_4$ + $FeSO_4$, reaching 57.8% recovery. It allows us to infer that adding both reagents is necessary to increase copper recovery. Furthermore, the analysis of residues and effluents indicates that the copper wad is the main mineralogical species that benefits from adding reducing agents.
- Black oxides have lower recovery and slower kinetics than traditional copper oxide ores. Thus, the time required to maximize recovery should be greater than 65 days of irrigation. Furthermore, according to the kinetic study, the curves of all the columns did not achieve maximum copper dissolution at the end of the leaching cycle.
- Finally, the present study of the leaching of the exotic mineral has allowed for satisfactory results for correct decision-making at an industrial level at CODELCO Salvador.

**Author Contributions:** Conceptualization, R.S. and M.M.; methodology, R.S., J.C. and M.M.; validation, P.H. and J.C.; formal analysis, A.G.; investigation, R.S. and M.M.; data curation, R.S.; writing—original draft preparation, M.M. and J.C.; writing—review and editing, P.H. and A.G.; visualization, J.C.; supervision, R.S. and J.C. All authors have read and agreed to the published version of the manuscript.

**Funding:** This research received no external funding.

**Data Availability Statement:** Data are contained within the article.

**Acknowledgments:** The authors wish to acknowledge that the language review was supported by the "FIUDA 2030" project. Additionally, Melissa Martinez thanks the Salvador Division of CODELCO for the support received and the University of Atacama, through the Vicerrectoría de Investigación y Postgrado (VRIP), for the postgraduate scholarship.

**Conflicts of Interest:** Melissa Martínez was employed by the company CODELCO. The remaining authors declare that the research was conducted in the absence of any commercial or financial relationships that could be construed as a potential conflict of interest.

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
