# Peer review of "Effect of Chloride and Ferrous Ions on Improving Copper Leaching from Black Copper Ores"

_processes, doi:10.3390/pr12010013_

Round 1
Reviewer 1 Report
Comments and Suggestions for Authors
Comments to the authors:
1. What was the efficiency of your study? Was 57.8% of leached copper amount obtained in both PLS + solid residues or only in PLS? Please clarify.
2. From the economic side of view is the method cost effective for such low-grade ore leaching?
3. M&M Section 2.1: Why did authors choose a size 12.7mm of copper ore? It is reported in the literature that the smaller the size as higher the dissolution. Did the authors study the effect of particle size on the leaching efficiency in their previous works?
4. Please specify at what temperature the Iso-pH test was performed.
5. Please specify at what temperature leaching column tests were performed.
6. Line 179-188: The authors reported literature results where the curing time is higher than those of the authors used. Please also report the literature data with the same curing time.
7. Line 191: Please specify the industrial operating parameters used.
8. Please mention the pulp density (PD) of copper ore in all tests used.
9. What about the Eh during the leaching in column tests? the dissolution rate is dependent on the redox potential. Did the authors record it or not?
10. Lines 206-208: According to the PLS results the authors reported that “…no significant differences are observed between the columns under study”. So, what was the purpose of pretreatment?
11. Line 222: Please in the Figure caption write “leaching”.
12. Line 229-231: The authors spoke about the stabilization of the
13. Cu2+/Cu+ pair by the effect of chloride ion and presented the reaction only for Cu2+, so please add the reaction for Cu+ as well.
14. Discussion section: In this section, the authors have to point out the novelty and advantage of the method they used in comparison to the existing ones.
15. Line 238-239: Did ≤2% of the difference of dissolved copper obtained via column leaching tests prove that the addition of FeSO4 was essential? Please rephrase the sentence.
16. Will be interesting if the authors can provide an X-ray diffractogram for solid residue.
17. Did the authors calculate standard errors? If yes, how?
18. References are quite less. Please add some more and if possible recent once.
Author Response
Dear Reviewer,
The corrections were answered in the attached file.
Thank you very much for your comments.
Kind regards
Rossana Sepúlveda

Reviewer 2 Report
Comments and Suggestions for Authors
Author Response

(The authors gave the same response as above.)

Reviewer 3 Report
Comments and Suggestions for Authors
This manuscript design iso-pH, curing, and column experiments to investigate the optimal conditions of Cu leaching from black copper ores, with special regard to the effects of chloride and Fe2+. Results showed that NaCl and reductant (FeSO4) can promote Cu release from black copper ores. This study has great industrial value in hydrometallurgy. I advise this work to be published in “”Processes“ after a major revision.
1. ittle information on the efficiency and mechanism of chloride to copper leaching was present in the abstract.
2. Reference 12 should be placed after "…with cryptomelane and birnessite".
3. The sentence "The exhaustive geochemical analysis of ..."(line 67) has no correlation to the sentences that followed.
4. Line 106: Is sulfuric acid used for extracting soluble Cu? What is the concentration of sulfuric acid?
5. Table 1: Two contents of element Ca were given. What is the meaning of "soluble copper"?
6. Table 2: Standard deviations should be added.
7. Line 120: “Iso-pH” should be “”iso-pH“. Initial acid dosage is unknown.
8. Table 3: Please check the unit of acid dosage.
9. Lines 147-148: Information about experiment column is repeated.
10. It is observed from Fig. 2 that more Cu was leached from column 1 than other columns. This result contradicts to the conclusion that NaCl and FeSO4 are positive to Cu release.
11. Table 6: How to determine the concentrations Fe2+ and Fe3+ in PLS?
12. Line 232: In such leaching system, formation of ferric sulfate, such as jarosite, is possible. Please check it.
Comments on the Quality of English Language
A comprensive language revision is needed.
Author Response

(The authors gave the same response as above.)

Reviewer 4 Report
Comments and Suggestions for Authors
Manuscript ID: processes-2705828
Title: Effect of chloride and ferrous ions to improve copper leaching from black copper ores
Authors: Rossana Sepúlveda et al.
Dear authors, my remarks will be in a very prosaic, businesslike style, however, please do introduce each of them with a "please". I am an advocate of directness in my comments, and I apologize in advance if any of them seem too harsh.
Line 86, 87. Avoid more than 3 references for a fact. A maximum of 3 in a sentence is allowed for Processes. Describe this information in detail.
Introduction. The authors have not provided detailed values for the processes described. It is necessary to add more values: technological parameters, metal recovery rates, etc.
Table 2. Add full chemical formulas of mineral phases.
Figure 1. Use colors for curves and Y-axis.
Figure 2-3. Use different colors for curves.
Author Response

(The authors gave the same response as above.)

Round 2
Reviewer 1 Report
Comments and Suggestions for Authors
Dear authors, thank you for taking into consideration all reviewer's comments and significantly improving the manuscript.
Reviewer 2 Report
Comments and Suggestions for Authors
no